# Visuotactile Affordances for Cloth Manipulation with Local Control

**Neha Sunil**[*1], **Shaoxiong Wang**[*1], **Yu She**[2], **Edward Adelson**[1], **Alberto Rodriguez**[1]
[1]Massachusetts Institute of Technology [2]Purdue University
<nsunil, wang_sx, albertor>@mit.edu shey@purdue.edu adelson@csail.mit.edu

**Abstract:** Cloth in the real world is often crumpled, self-occluded, or folded in on itself such that key regions, such as corners, are not directly graspable, making manipulation difficult. We propose a system that leverages visual and tactile perception to unfold the cloth via grasping and sliding on edges. By doing so, the robot is able to grasp two adjacent corners, enabling subsequent manipulation tasks like folding or hanging. As components of this system, we develop tactile perception networks that classify whether an edge is grasped and estimate the pose of the edge. We use the edge classification network to supervise a visuotactile edge grasp affordance network that can grasp edges with a 90% success rate. Once an edge is grasped, we demonstrate that the robot can slide along the cloth to the adjacent corner using tactile pose estimation/control in real time.

**Keywords:** Multi-modal learning, Cloth manipulation, Tactile control

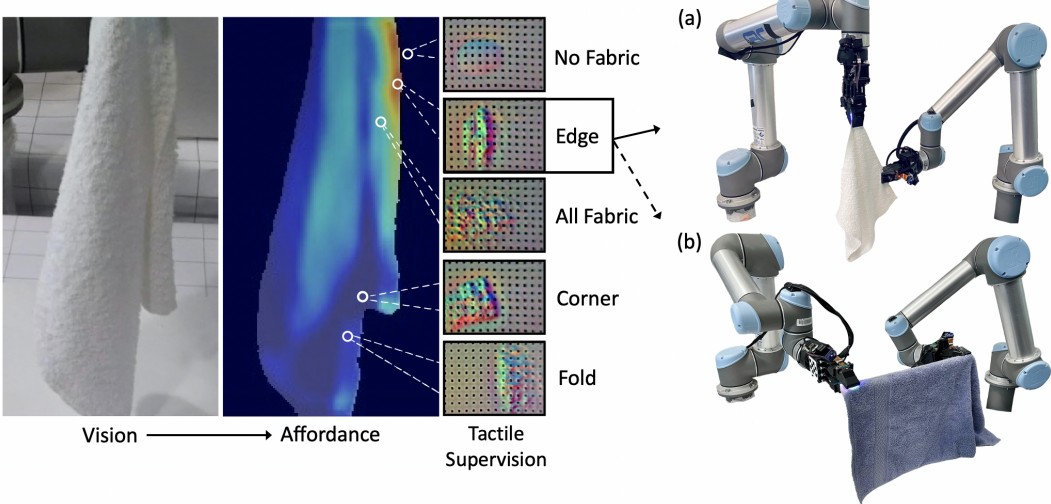

Figure 1: **Visuotactile affordance for sliding.** Using the visual depth image as input, our affordance network scores each pixel in terms of potential edge grasp success. The affordance image shown is trained first in simulation and then fine-tuned with tactile self-supervision from real grasps. In this particular cloth configuration, the adjacent corner is not directly visible or graspable, motivating first grasping an edge and then sliding. We demonstrate tactile sliding in two configurations: (a) sliding down with the cloth in a stable configuration and (b) horizontal sliding suitable for longer towels.

## 1 Introduction

The robotic manipulation of highly deformable objects is a growing research area due to its applicability in various fields such as assembling cable harnesses in factories, industrial garment manufacturing, and assisted dressing and laundry folding in the home and healthcare industry. Our goal is a

---

[*]Authors with equal contribution.
This work was done at MIT.

6th Conference on Robot Learning (CoRL 2022), Auckland, New Zealand.

framework that combines visual and tactile sensing of local features (specifically cloth corners and edges in our system) for perception and control of deformable objects. Within this paper, we focus on the manipulation of cloth.

Manipulating deformable objects is difficult due to their continuous and highly-underactuated nature. Applicable high-dimensional models are challenging to identify or simulate. Perception is another key challenge, given that the object is easily occluded by the robot's grippers and even self-occluded by the object itself. These issues pose an inherent difficulty in object representation since full state estimation and modeling is very challenging for real-time manipulation.

Most cloth manipulation studies address these challenges with visual perception [1]. A common first step is unfolding the cloth to make specific features visible by holding it up or flattening it on a table. Then, the robot can perceive and directly grasp features (like corners or collars) for subsequent manipulation tasks. However, these features may still not be directly graspable, as in the corner example in Fig. 1.

In contrast, we want to approach real-time, closed-loop manipulation that is closer to how humans manipulate deformable objects. Instead of directly looking for adjacent corners on a crumpled cloth to pick it up along an edge, we unfold the cloth while sliding until we feel a corner. As shown in Fig. 1, once an edge is successfully grasped, we can slide to the adjacent corner, which was not otherwise graspable. This bottom-up approach of using local features (specifically corners and edges) for perception and control, enables novel manipulation skills.

We leverage tactile perception, along with vision, to localize and manipulate local features. Vision provides global guidance to help with state identification and grasp point localization (in the style of visual affordance-based grasping [2, 3]). We introduce visuotactile affordances where we learn grasp affordances self-supervised with tactile feedback. Tactile feedback from the gripper aids in the classification and estimation of the state of local features. This is especially useful when vision is occluded by the gripper. Finally, we use tactile perception for grasp adjustments, local control, and grasp success confirmation. In this paper, the key technical contributions are:

- **Tactile Perception of Local Features.** We demonstrate a strategy for local feature pose estimation and classification where we show that it is possible to distinguish between edge, corner, fold, all fabric, and no fabric grasps with 92% accuracy (Sec. 4.1).
- **Visuotactile Grasp Affordances.** A grasp localization affordance network pre-trained in simulation and fine-tuned with tactile supervision with a 90% grasp success rate (Sec. 4.2).
- **Tactile Sliding.** Tactile based controllers to slide a gripper along the edge of cloth in two different configurations (Sec. 4.3).

Ultimately, our key experimental contribution is a demonstration of our framework picking a crumpled towel from the ground, grasping a corner using vision, grasping an edge using our visuotactile affordance network, and sliding to an adjacent corner, reaching a configuration which can be used to fold or unfold the towel.

## 2 Related Work

Perceiving the full state of cloth [4, 5, 6] or modeling its dynamics [7, 8, 9] are challenging and time-consuming approaches. In this section, we instead highlight works that grasp and manipulate local features for cloth folding and unfolding.

**Manipulation of Local Features of Deformable Objects.** Much of the work on cloth-centric perception for manipulation focuses on corner detection and ridge detection for determining grasp points for future manipulation tasks [10]. Both can be done using computer vision techniques, often using filters (e.g. Harris corner detection [11] and Gabor filters for wrinkles [12]). However, finding other more specific local features often requires first flattening the cloth [13, 14, 15, 16] or hanging it from specific grasp points [17, 18, 19] to prevent self-occlusion of the cloth.

Learning-based perception approaches enable more flexibility with different local features and starting configurations. However, determining ground truth labels is challenging for cloth. Qian et al. (2020) trains a network to segment edges and corners of cloth from a depth image by using a painted cloth as a ground truth label [20]. Simulators can also provide these labels [21, 22], but Sim2Real transfer is an issue, given the uncertain dynamics of cloth. In this work, we first train our edge grasp affordance network in simulation and fine-tune on the robot using tactile perception for supervision.

Most work in fabric manipulation either uses incremental pick-and-place movements against a table [14, 23, 21] or holds up the cloth and regrasps while hanging [17, 18, 19]. However, the sliding skill simplifies the task of finding two adjacent corners in order to fold a piece of fabric, especially when the second corner is not immediately visible or graspable. Sahari et al. (2010) holds up a corner of the fabric and uses gravity to trace downward to find the second corner with low-resolution sensory feedback [24]. Similarly, Yuba et al. (2017) executes an open-loop "pinch and slide" motion along the top edge of a piece of fabric [25]. Zhou et al. (2020) use low resolution tactile sensing to learn a sliding policy in latent space [26]. Our high-resolution tactile sliding allows us to consistently follow longer towels while being robust to disturbances in both stable and horizontal configurations.

In previous work, Yu et al. (2019) slides along cables using GelSight [27] for tactile perception to estimate the pose of the cable in grasp while sliding [28]. Gelsight, and other vision-based tactile sensors, convert touch to vision by using a camera to visualize the deformation of the contact surface. We also approximate shear force by tracking the black markers on the sensor surface as a proxy for friction force while sliding.

**Perception of Local Features.** Visual grasp affordances are proposed for learning grasping configurations with data-driven methods [3]. The affordance is a metric to predict the grasp success rate under different configurations. Each configuration is represented as an individual pixel and a discrete rotation angle. Affordances can be labeled manually [3], but alternatively can be learned in a self-supervised manner [29, 2, 30, 16]. In these cases, the supervision for affordances comes from either simulation or the visual outcome on a real system, which usually requires the end-effector to move out of the camera frame for visual evaluation. In contrast, our visuotactile approach fine-tunes the affordance trained in simulation under visual occlusion. The learned visuotactile affordance allows for a smooth transition between grasping and sliding.

## 3    Cloth Manipulation

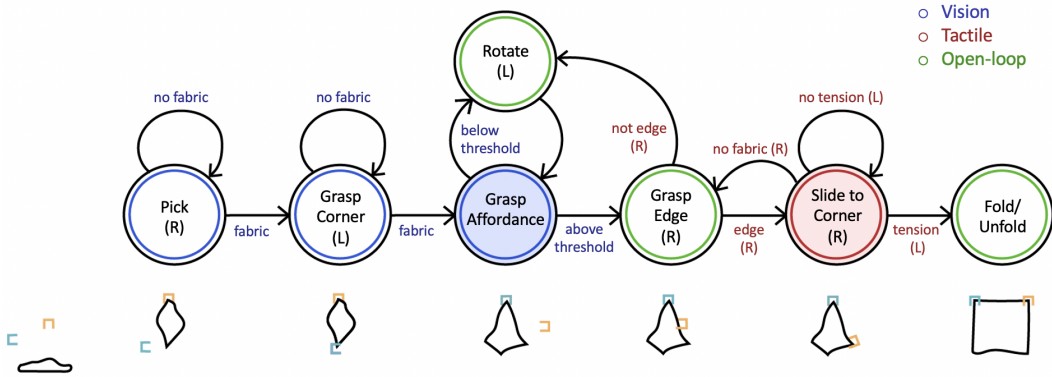

Figure 2: **State machine for cloth manipulation.** Colors in the state machine reflect sensing modalities required for each action and transition conditions between actions. The two key components, visuotactile grasp affordances (highlighted blue) and tactile sliding to corner (highlighted red), are discussed in Sec. 4.2 and Sec. 4.3 respectively. Below, we illustrate the cloth state with respect to each gripper.

In order to fold or unfold the cloth, we first pick the towel up from a surface, grasp a corner, grasp an edge adjacent to that corner, and slide to the adjacent corner (See Fig. 2). Once the robot is grasping two corners, it can complete the folding or unfolding task open-loop.

**Tactile Sliding.** In the cloth sliding task, two grippers start near each other, grasping the cloth edge. The goal is to traverse as much of the fabric edge as possible or to a specified goal condition (a corner, in this work) without losing the grasp, maintaining an estimate of the edge pose within the sliding gripper.

Sliding along the edge of short towels can be done with regrasps if the cloth is kept in a stable configuration under gravity. We use a simple control strategy for sliding along a cloth edge while hanging. However, a longer towel is more likely to be entangled, susceptible to disturbances, and partially resting on the table. In addition, the adjacent corner is likely self-occluded, making the sliding skill more useful. Thus, we extend the work of Yu et al. (2019) for sliding along cables to longer towels [28]. Here, a moving gripper with a fixed grip on the cloth pulls the cloth through a stationary gripper while attempting to keep the edge visible in the tactile image of the stationary gripper, as shown in Fig. 1. Both sliding strategies are discussed in Sec. 4.3. Perception of the cloth edge is more difficult than for cables. Therefore, we use a supervised data-driven method to estimate the pose of the fabric edge, explained in Sec. 4.1.

**Grasp Affordance for Sliding.** In order to grasp an edge in a way that allows for sliding, a single layer of fabric should be aligned within the gripper to avoid collisions. We train a visuotactile grasp affordance network that takes the depth image as input and outputs an affordance heatmap, indicating the grasp viability of each point on the image. This network is originally trained in simulation. Then, we fine-tune this network on a robot using the tactile information from the grasp attempt as supervision (Sec. 4.2).

## 4 Method

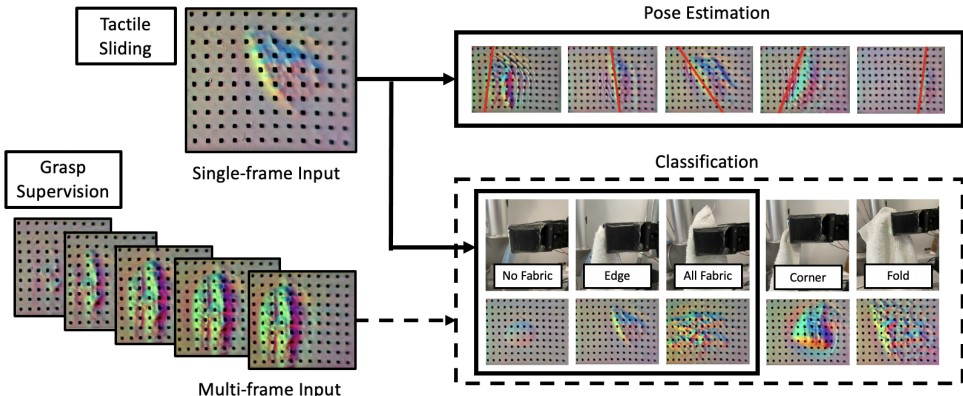

Figure 3: **Tactile perception.** The perception network used for sliding along the fabric edge takes a single-frame input and outputs the pose of the edge, including states where no edge is visible (the no fabric and all fabric cases). For our grasp supervision network, we sample multiple frames from the grasp attempt and classify between additional categories.

### 4.1 Tactile Perception of Local Cloth Features

Tactile perception is used for pose estimation and classification of local features within the grip. All of the following tactile perception networks share a similar architecture and strategies for data augmentation.

**Pose Estimation.** Applying the same cable pose estimation method as [28] to slide along the edge of soft cloths would be challenging because the fabric is relatively thin and covers a large area (see Fig. 3), unlike the cable which creates a distinct imprint. We found that traditional computer vision edge detection techniques are unsuccessful at consistently isolating the fabric edge due to the fabric texture, wrinkles in the sensor's gel surface, and noise in the tactile image.

Our pose network takes a down-sampled depth image as input and outputs the x and y coordinates of the center of the edge and the orientation of the edge, which can be directly used for tactile sliding.

The network has 3 convolutional layers followed by max pooling, and 3 fully connected layers. We choose a simple network architecture to optimize for speed to use in our sliding controller.

The dataset is generated from a small number of hand-labeled tactile images (frames from a video), and then augmenting the dataset by randomly varying the maximum depth threshold of each tactile image, and further augmenting the images by introducing a random translation and rotation (see Appendix for details). The human labeling process involves clicking two points to define the cloth edge if visible, or otherwise classifying the image as all fabric or no fabric. This labeling can then be transformed to all the synthetically augmented images.

**Classification.** The tactile sliding perception network that outputs pose also classifies whether (1) there is no fabric on the sensor, (2) the sensor is covered in fabric, or (3) the sensor sees an edge (Fig. 3). We collect data from videos of sliding with different gripper widths. The final network we train has 150 raw training images which we augment to 30,000 depth images.

The tactile sliding network can classify between no fabric, edges, and all fabric given a single frame when the gripper is pressing against the towel. To supervise our edge grasp affordance network, we also need to distinguish between edges and folds. Fold classification is especially challenging because the final tactile image can look identical to other categories. Therefore, we use a sequence of images from the grasp attempt as input which introduces temporal information, instead of just the final grasp. The tactile image starts showing an imprint earlier with multiple layers of fabric.

We sample every 5 tactile depth images of the grasp attempt and choose the last 5 as input to the network. During the augmentation process, we shift the initial frame for sampling. We skip depth augmentation because timing is crucial to this network, but we continue to translate and rotate the images. We record 330 grasp attempts per category and augment this to over 6000 per category.

## 4.2 Visuotactile Grasp Affordances

In this work, there are three kinds of grasp points we are interested in: generic (for picking), corner, and edge. All three use vision to choose grasp points and can use either visual or tactile perception to verify grasp success (Fig. 2). We discuss the generic and corner grasps in Sec. 4.4.

Edge grasp point localization is more challenging than corners because we must also consider the direction of the edge. Furthermore, visual edge detection can be difficult since edges look similar to folds along the contour of the cloth. The cloth edge can also easily twist in on itself, but a successful grasp for sliding requires grasping a single layer. Moreover, aligning the edge within the gripper to avoid collision is crucial for grasp success. If the cloth collides with the gripper, it easily deforms, preventing a successful grasp. Given these additional considerations, we train a network that learns visuotactile edge grasp affordances.

We first train this affordance network in simulation. Then, we refine our fine-tuning technique while transferring between slightly different simulation environments. Finally, we fine-tune this network on a robot using the tactile information from the grasp attempt's success/failure as supervision.

**Training in Simulation.** We use the same Blender simulation setup as [21] for training the affordance network using U-Net [31], as shown in Fig. 4. The input is a depth image of the hanging cloth, and the output is the affordance heatmap. The ground-truth

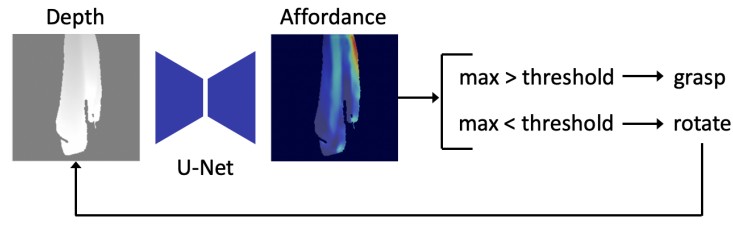

Figure 4: **Affordance architecture and rotation policy.**

labels of the affordances in simulation are determined for each pixel by geometric computing, specifically (1) the percentage of neighboring cloth nodes that are classified as an edge, (2) whether the

gripper will collide with cloth, (3) whether the gripper would grasp a single or multiple layers of fabric, and (4) whether the point is reachable. All of these criteria can be determined in simulation because we have access to the full state of the cloth. The tactile sensor implicitly checks all of these criteria on the real system when it classifies a grasp as an edge or non-edge grasp. The simulated dataset contains 200 cloth configurations rotated in increments of $15°$, yielding 4800 depth images with corresponding affordance images.

**Fine-tuning in Simulation.** As a precursor to sim-to-real transfer, we first try sim-to-sim transfer to validate our network architecture and tune parameters. While the network in simulation has access to an entire labeled image during training, during fine-tuning, the robot can only test a single pixel at a time. We create a second simulated dataset with slightly different camera location, cloth size, and cloth stiffness. We change several network parameters (see Appendix for details) and ultimately find that fine-tuning with single pixel updates is feasible with an experience replay buffer.

**Fine-tuning on Robot with Tactile Supervision.** The real depth images on the robot have regions of zero-depth due to the offset in location between the projector and sensor of the depth camera. Also, items in the background affect the affordance. Therefore, we mask out all non-cloth regions in both simulation and deployment and add random black rectangles to the simulated depth images during training for robustness against regions with zero-depth (see Appendix for details).

During data collection, we grasp a corner, mask the reachable regions, and attempt to grasp the cloth at different points with fixed orientation before resetting and grabbing a new corner. The reachability mask avoids collisions between both grippers, chooses the side of the cloth closest to the grasping gripper, and filters out the bottom of the cloth to only grasp adjacent edges. The cloth is rotated in increments of $15°$ for a whole rotation with 3 grasp attempts per rotation. We implement a minimum affordance threshold to increase the number of positive examples (Fig. 4). Each grasp attempt is labeled with the tactile multi-frame classification network's confidence that the grasp is an edge. We collected around 3,000 grasps, which takes about 15 hours. We validate the network with 110 hand-labeled data collected offline.

### 4.3   Tactile Sliding

**Sliding in Stable Configuration.** Once one gripper holds a corner and the other holds an adjacent edge, we use the tactile signal to slide downward. The sliding gripper moves forward and back to keep the edge close to the target position within the grip until the robot reaches a corner. We use a proportional controller with respect to the fabric edge's offset from the target position to slide along the edge. We consider an increase in shear (as determined by GelSight marker displacement) above a certain threshold from the fixed gripper to indicate that the sliding gripper has reached the thicker corner.

**Sliding along Long Towels.** We follow the methods of [28] to slide along the edge of longer towels in a horizontal configuration. We learn a linear dynamics model that finds the relationship between state (defined by cloth edge position and orientation and robot position) and pulling angle. We use this model in an LQR controller that keeps the edge close to the target state as the cloth is pulled through (see Appendix for details).

### 4.4   Cloth Manipulation System

In order to complete the system, we also need to pick the towel up from the ground and then grasp a corner. Although the specific grasp location with respect to the rest of the cloth is unknown, this initial grasp is helpful for partially unfolding the towel using gravity. This allows for more of the towel to be visible for downstream tasks. For our specific grippers, the towel grasp point cannot be completely flat on the surface (wrinkles are much easier to grasp). Therefore, we sample from the highest points to choose a random grasp point.

Corner grasp points are common for flattening and folding tasks. However, corners are not always visible when the cloth is on a surface due to self-occlusions. Furthermore, visible corners on a table often lie flat on the surface, which would require modifications to our gripper design to grasp them. So, instead, we look for corners on the hanging cloth by choosing the lowest point.

For the hardware setup, we use dual UR5 robot arms, which are equipped with parallel-jaw grippers servoed with dynamixel motors (Dynamixel XM430-W210-T, Robotis). We mount two GelSight sensors [32], one on each gripper. We use a Kinect Azure camera to capture RGB-D images.

## 5 Results

**Tactile Perception.** Our pose estimation network for tactile sliding has a mean average error of 3.2 mm for the center position of the edge and $4.1°$ for the orientation of the edge. The overall sensing area is 30 mm wide. These error ranges are within the expected human labeling error and this performance is experimentally sufficient for tactile sliding. We compare this network to one trained on a dataset without augmentation, which has a similar position error, but a significantly larger orientation error of $10.2°$. This network runs at 33 Hz within the sliding controller.

The multi-frame classification network used for grasp affordance supervision has an overall classification accuracy of 92%. Since our affordance network only needs to distinguish edges from other grasps, we also report a non-edge classification accuracy of 98%.

**Visuotactile Grasp Affordance.** Our network trained in simulation successfully grasps edges in 21.3% of the given configurations, out of which only 39.7% have valid, accessible edge grasps. By introducing our threshold rotation policy (Fig. 4), our network successfully grasps an edge for 56.6% of initial configurations in which 95.0% of configurations had valid grasps. Rotations are crucial to finding and accessing valid edge grasps.

We compare our fine-tuned affordance network to three other baselines: (1) Segmentation-based: a visual edge segmentation network [20] combined with surface normal and right side-reachability cost metrics (see Appendix for details), (2) Sim2Real: the network trained in simulation that is directly transferred to the robot, and (3) Real2Real: a network trained directly on the collected robot data. We evaluate the networks offline using precision@k [33], using this metric due to the unbalanced nature of the evaluation set and because it does not need a set threshold. We found that Sim2Real performs the worst because of the differences between the simulated and real robot environments. However, the fine-tuned network outperforms the Real2Real network, showing that the network from simulation provides overall structure.

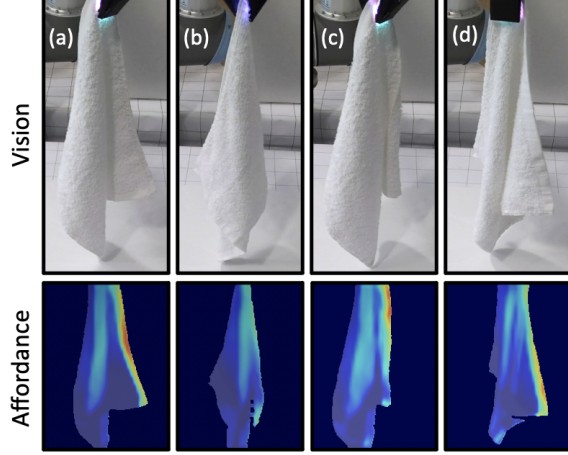

Figure 5: **Fine-tuned edge grasp affordance results.** The learned affordances effectively reflect the edge graspability in different configurations, where it is graspable for (a) the entire right edge and (b) no edge. The last two towels have edges that twist such that they are only accessible from (c) the top portion and (d) the bottom portion. The proximity to the rest of the fabric (c) or the misalignment of the surface normal (d) would cause collisions during the grasp attempt.

We further evaluate the networks in deployment by reporting the average number of grasp attempts before identifying an edge (as confirmed by tactile perception) and the overall success rate of true edge grasping. The threshold was chosen independently for each network and tuned to maximize performance on the offline dataset. The cloth is grasped by the corners and evaluated with 10 different configurations.

|  | Offline Evaluation | Real Robot Deployment | |
|  | Precision@k | Average # attempts | Success Rate |
|---|---|---|---|
| Segmentation-based [20] | 65.0% | 4.2 | 65.0% |
| Sim2Real | 45.0% | 1.3 | 40.0% |
| Real2Real | 62.5% | 3.3 | 50.0% |
| **Sim2Real + Finetune (Ours)** | **80.0%** | 1.6 | **90.0%** |

Table 1: Quantative results in real world for visual affordance learning with tactile supervision. We use $k = 40$ for precision@k, since there are around 40 positive samples out of 110 in the real-world data for offline evaluation. For evaluation during deployment, we compare methods with 10 different trials. Each model has similar initial cloth configurations for each trial.

Table 1 shows the experiment results. The failure modes include false-positive edge tactile classifications and rotating $360°$. In general, more grasp attempts increase the likelihood of a false positive edge classification. Our fine-tuned network outperforms both baselines with an overall success rate of 90%. The Sim2Real network often misses grasp opportunities and completes a full rotation without successful attempts. The Real2Real network is more likely to grab inside the cloth without the underlying structure from simulation. The robot often grabs an edge along with a fold, resulting in false positives from tactile classification. The segmentation-based network fails to identify grasps that would fail due to collisions with other regions of fabric. The fine-tuned network effectively identifies graspable edges, considering both global and local geometry and semantics for calculating affordance (Fig. 5).

**Tactile Sliding.** When sliding while the towel is in a stable configuration, we find that our sliding controller performs differently with the thin and thicker edges of the towel. The gripper traverses 100% of the thin edge (sliding all the way to the corner) no matter the initial edge position and 100% of both the thin and thicker edges if the initial edge pose is near the inner edge of the gripper (n = 5 trials). The sliding distance decreases (cloth slips from grip before reaching the corner) as the initial edge pose starts closer to the fingertip. At a centered initial pose, the sliding gripper travels an average of 95% of the towel length (14.0 cm for the thin edge and 12.7 cm for the thick edge).

For sliding in the horizontal configuration, we slide until the end of the robot workspace (smaller than the towel's total length). Similar to sliding in a stable configuration, we find that if the cloth edge starts near the gripper's inner edge, the robot can consistently slide for the entire length of the workspace in all trials. If the initial edge pose is in the center of the gripper, the robot travels an average of 76% of the workspace (43 cm out of 56 cm).

## 6 Discussion

We develop a robot system for cloth manipulation that uses both visual and tactile perception of local features for local control. Our tactile perception systems are used for both pose estimation for controlled tactile sliding and supervision for learning visuotactile grasp affordances. Both networks are trained with small amounts of real data by augmenting the tactile depth images. Our grasp affordance network learns global structure from simulation and is fine-tuned using tactile supervision during real robot grasp attempts. This network outperforms all of our baselines.

**Limitations and Future Works.** We use multiple neural networks for our local approach to perception and control, which lends itself to generalization because we do not rely on the global structure of the deformable object. However, while our pose network works on most towels, but the tactile classification network is specific to the training towel's thickness and texture and the fine-tuned affordance network is specific to the towel's dynamics and shape. These networks could be more generalizable if trained with a dataset with more variety. Our fine-tuning framework allows for introducing new cloth types without needing to train from scratch (given an applicable classification network). Our tactile approach allows certain modules to be independent of specific cloth parameters, for example, towel length or texture while sliding. As another limitation, tactile sliding success is dependent on the initial cloth edge position. Adding a tactile regrasp step to adjust the cloth edge position with tactile feedback before sliding could help address this issue.

## Acknowledgments

We thank the anonymous reviewers for their helpful comments in revising the paper. Toyota Research Institute (TRI), Amazon Research Awards (ARA), the Google Faculty Research Awards, the Office of Naval Research (ONR) [N00014-18-1-2815], and GentleMAN and BiFrost projects of SINTEF provided funds to support this work. Neha Sunil is supported by the National Science Foundation Graduate Research Fellowship [NSF-1122374].

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
