# OpenReview forum: "Visuotactile Affordances for Cloth Manipulation with Local Control"
_robot-learning.org/CoRL/2022/Conference — CoRL 2022 Poster_

### Official Review · Reviewer_6ikq · 2022-07-27

**Originality:** Good
**Technical Quality:** Good
**Clarity Of Presentation:** Very Good
**Impact:** 4

**Recommendation:**

Weak Accept: I recommend accepting the paper, but will not argue for my recommendation if the majority of other reviewers have a different opinion.

**Summary:**

This paper presents a system that uses vision and tactile perception to unfold clothes by grasping and sliding. The system uses two UR arms, depth camera and vision-based tactile sensors attached to the robots' grippers. Perception is handled with neural networks that are responsible for the classification of grasping edges, supervision of visuo-tactile affordances detection, and pose-estimation/control for sliding. The system is deployed on real robots effectively.

**Issues:**

Missing or limited details: please add implementation details for reproducibility, as well as quantitative measures indicating uncertainty/effectiveness in your experiments, for example prediction/classification uncertainty, robustness to noise/disturbances, reliability/repeatability of the perception/control loop.

**Quality Of The Limitations Section:**

Additional details required

**Reviewer Expertise:**

3: The reviewer is fairly confident that the evaluation is correct

**Robotics Focus:**

Sufficient demonstration on hardware

**Strengths And Weaknesses:**

The paper presents advances on a relevant and challenging topic in robot manipulation. The problem of cloth manipulation involves several challenges, that are addressed in a systematic and clear way in the proposed work.
The paper is well structured, it is clear and easy to read. Challenges and contributions are presented in a clear way, and the literature review seems comprehensive to the best of my knowledge. Figures, diagrams and videos are good and helpful to get a grasp of the contributions.
The experiments include an informative set of comparisons, and the relative discussion includes useful insights and analysis.
The limitations reported point out relevant shortcoming of the proposed method.

While the paper explains the experimental setup and method thoroughly, implementation details are missing: for example details related to the networks trained and used in the experiments, how many parameters were used, how long the training time is, etc. Some information on how the robot is controlled, and how the state machine (fig.2) is deployed would be very useful as well to provide a complete picture of the experimental setup.
The analysis would benefit from a discussion on what the failure modes are and why they happen. For example, what are the limitations on the perception side? do you have a quantitative measure of the uncertainty of the tactile classifier? how does that affect the final performance? on the visual perception, what would make your setup more robust?
The sliding motion seems really hard to obtain, and the results presented are remarkable. Do you have quantitative measures of the robustness or reliability of this controlled movement? How many repetitions are consistently executed on the same piece of cloth? What part of your setup/architecture would further improve the overall performance?




**Summary Of Recommendation:**

The paper presents a relevant piece of work, that tackles a challenging and important problem in robot manipulation. The execution of the experiments and the research advances are interesting and, although not general, very promising.
While some implementation details are missing, the work is well executed and the experimental results are remarkable.

---

> ### Author Response · Authors · 2022-08-28
> **Response to Reviewer 6ikq**
>
> We thank reviewer 6ikq for their helpful feedback. We address their concerns as follows and added details to the appendix in the following sections:
>
> **“details related to the networks trained and used in the experiments"**
> We have added a Network Architecture and Training Details section to the appendix (7.3):
> “We choose a small network architecture for our tactile sliding network to optimize for speed. Specifically, our base structure is C6-C16-C32-MP-F512-F128... "
>
> **Some information on how the robot is controlled**
> We have added a Robot Implementation and Control section to the appendix (7.8):
> “Grasping with the UR5s uses position-control while tactile sliding uses velocity-control for smooth behavior, which runs at around 30 Hz. For the grippers, we use Dynamixel motors. They are position-controlled with current limits to protect the sensor...”
>
> **“how the state machine (fig.2) is deployed would be very useful as well to provide a complete picture of the experimental setup”**
> We have added details about the state machine to the System Design Choices section in the appendix (7.1):
> “In the state machine, the robot tries to pick up the towel from the ground with its right gripper, then grasp a corner (the lowest point of the towel) with its left gripper. After each of these steps, visual perception is used to ensure that the grasp is successful, meaning that there is cloth in the expected region of the depth image. If at any point the towel is on the ground, then the state machine restarts from the initial pick. Once a corner is grasped, the robot uses our grasp affordance network to find an edge grasp whose affordance value is above a certain threshold. If no grasp is found, the towel is rotated. If a full rotation is reached, then the towel is released to the floor and the state machine restarts. The gripper approaches the edge with a fixed orientation. After grasping an edge (confirmed using tactile), the gripper slides to the adjacent corner, where the end condition is determined from tactile...”
>
> **The analysis would benefit from a discussion on what the failure modes are and why they happen. For example, what are the limitations on the perception side?**
> We have added a Failure Modes section to the appendix (7.2) discussing corner grasping, edge grasping, and the grasping to sliding transition:
> “The main failure modes of corner grasping are slightly missing the towel (because of movement while sensing or errors from the camera) or the lowest point not being a corner (if the towel is partially folded)...”
> “Edge grasping fails if the tactile classifier falsely determines an edge is grasped (especially if grasping a compressed fold). Another common failure mode is that the towel is rotated fully without a viable edge grasp. Sometimes, the edge is visible but sticks to another layer of cloth...”
> “The edge grasping to sliding transition is difficult because some successful edge grasps can be challenging for sliding. The position of the edge is not a consideration for the edge grasp network, while it is a factor that highly influences sliding success...”
>
> **"do you have a quantitative measure of the uncertainty of the tactile classifier?"**
> We have added a Tactile Grasp Supervision Network section to the appendix with a confusion matrix (7.5):
> “Tactile classification accuracy is evaluated on a held out test set evenly distributed across classes. We use a network that best differentiates between edge and non-edge grasps. The network performs well on most categories except for folds, which can look like most other categories. However, the network is sufficiently capable of distinguishing edges and folds for the purposes of our grasp supervision network.”
> Tactile misclassifications can lead to false positives for edge confirmations, resulting in the robot sliding along a fold.
>
> **"on the visual perception, what would make your setup more robust?"**
> We could use multiple cameras with different viewpoints to help differentiate between edges and folds and to ensure that edges are easy to separate from the rest of the cloth. In terms of generality of the solution, the system would benefit from a global visual prior of the structure of the cloth to inform the manipulation of more complex pieces of cloth or more complex folding tasks.
>
> **"The sliding motion seems really hard to obtain, and the results presented are remarkable. Do you have quantitative measures of the robustness or reliability of this controlled movement?"**
> We have added an experimental results table and more details on our evaluation of vertical tactile sliding to the appendix (7.9):
> “To test the robustness of vertical tactile sliding, we varied the initial edge position across experiments with 5 trials per initial condition (Table 2)...”

---

### Official Review · Reviewer_MA9Y · 2022-07-28

**Originality:** Very Good
**Technical Quality:** Very Good
**Clarity Of Presentation:** Excellent
**Impact:** 4

**Recommendation:**

Strong Accept: I recommend accepting the paper and will argue for my recommendation even if other reviewers hold a different opinion.

**Summary:**

This paper develops a manipulation framework for picking up and unfurling cloth in tabletop scenarios (e.g. cloth randomly draped on table). The system works by (1) picking up cloth from the table; (2) re-positioning the cloth so it can be grasped at a (detected) corner, and (3) unfurling the cloth by “following” an edge from the detected corner. The authors found that tactile data (from GelSight) was crucial both as an **input** *and* as a **supervision source** for training the necessary perception models (e.g. corner/edge detectors). Data for training these models was collected both in simulation and in the real world. The authors validate these design choices with an ablation study, and demonstrate qualitative behavior in their video.


**Issues:**

The authors should address the weakness pointed out above, by giving a clearer idea of how their method improves end-task (i.e. cloth unfolding) success rate in realistic scenarios.


**Quality Of The Limitations Section:**

Limitations are addressed clearly

**Reviewer Expertise:**

3: The reviewer is fairly confident that the evaluation is correct

**Robotics Focus:**

Sufficient demonstration on hardware

**Strengths And Weaknesses:**

**Strengths**
* This paper offers a compelling examples for how to use tactile information to solve real world robotics problems. It is particularly interesting that the touch data is used to improve the vision model through supervision, as well as being used as an input.
* The authors clearly did some great task specific engineering and came up with clever strategies for unfolding.
* The video does a good job explaining the method and contextualizing results.

**Weaknesses**
* The experiments only discuss success rates in terms of affordance learning, but *not* in terms of task success. This makes it hard to contextualize their improvement over baselines for the final task.

**Summary Of Recommendation:**

This paper showcases an impressive framework for solving a challenging real world task. I particularly enjoyed reading how the authors exploited unique sensing capabilities (e.g. tactile data), and what was required to successfully transfer their perception stack from sim to real. While I would like more context on task success (see Issues section), I don’t believe this issue is major enough to warrant rejection.

---

> ### Author Response · Authors · 2022-08-28
> **Response to Reviewer MA9Y**
>
> We would like to thank reviewer MA9Y for their helpful feedback. We address their concerns as follows:
>
> **“The experiments only discuss success rates in terms of affordance learning, but not in terms of task success. This makes it hard to contextualize their improvement over baselines for the final task.”**
> The original intention of this paper was to show the capabilities of visuotactile perception and control, rather than overall task success/efficiency. While each skill has a high success rate if initialized properly, the transitions require more engineering to make smooth. As addressed in the limitation section, sliding success is highly dependent on initial edge position and orientation. To address this, we could use edge position as supervision or introduce a tactile regrasp. Also, whenever an action fails, our visual and tactile checks (if accurate) can restart the state machine where appropriate. We added a system design choices section to our appendix (7.1) to discuss this decision.

---

### Official Review · Reviewer_Bk2e · 2022-07-28

**Originality:** Good
**Technical Quality:** Very Good
**Clarity Of Presentation:** Good
**Impact:** 3

**Recommendation:**

Weak Accept: I recommend accepting the paper, but will not argue for my recommendation if the majority of other reviewers have a different opinion.

**Summary:**

This paper describes a method for manipulating a towel from its initial configuration to a configuration where adjacent corners are grasped, after which subsequent tasks could be performed like folding. Using a bimanual robot setup, they train a tactile classifier to distinguish between edges, folds, and non-edges to confirm successful edge grasps. A "tactile affordance network", trained from synthetic depth images and fine-tuned in real, predicts viable edge grasps given a towel dangling from one gripper, and uses the above classification network to self-supervised collect 1000s of additional real datapoints. An additional output of the classification network is the position and orientation of the edge, used for closed-loop control during sliding.

In experiments, they test on (1) a small towel, and (2) a large towel, with different manipulation strategies for each. The system exhibits great performance on small towels, and performs reasonably on larger, more challenging towels.


**Issues:**

I think the writing changes suggested in strengths/weaknesses should be addressed, and experiments should be run with full pipeline rollouts folding small towels from an initial crumpled state to report overall success rate on the goal task.  However I understand the difficulties of adapting the system to conduct full pipeline rollouts with large towels, so this aspect is less important during revisions, although would significantly strengthen the paper if included!

**Quality Of The Limitations Section:**

Limitations are addressed clearly

**Reviewer Expertise:**

4: The reviewer is confident but not absolutely certain that the evaluation is correct

**Robotics Focus:**

Sufficient demonstration on hardware

**Strengths And Weaknesses:**

Strengths:
- The approach proposed is a sensible, well-motivated strategy for interacting with large deformable 2D objects.
- The related work section is thorough and well contextualized.
- The methods presented are technically sound, well-executed, and feasible to scale to different types of cloth with additional work on data collection.

Overall I enjoyed the paper, and feel with some writing clarity changes and additional experiments it is a valuable contribution to the deformable object manipulation community.

Suggested changes:
In my view there are two key weaknesses that should be addressed, and some smaller comments which I left in a list below.
1. $\textbf{Writing clarity}$: The writing is unclear in a some places, specifically around experiment details and the construction/usage of the classification network(s).
- How was the threshold for grasp execution chosen?
- How is the orientation of an edge grasp determined from the pixel in the depth image selected via the affordance network?
- How is the 92% tactile classification accuracy evaluated? using a test set from the original data? What is the distribution of classes in the test set and how many of each?
- It seems to me that the corner classification from the tactile network is never used in the pipeline, and instead shear force serves as a proxy for detecting a corner while sliding. Why not use the available corner classification network for this part of the system? The experiments section states that corners are grasped by "choosing the lowest point" while dangling a towel, but it's unclear if the corner classification is used here to confirm a grasp, and what happens if a corner is not grasped. If it's never used, I would suggest removing it from fig 4 and the video/text for clarity.
- Fig 4 seems to conflict with the classification section: in the text it's stated that the tactile classification network distinguishes between {no fabric, all fabric, edge}, and the grasp supervision network can additionally distinguish between edges and folds, however corners are never mentioned. Fig 4 suggests that the grasp supervision network can detect corners
- The success rate of corner grasping (sec 4.4) is not discussed: is it 100%?
- The initialization of experiments is unclear: is the procedure described in sec 4.4 used to initiate every experiment? What happens if the corner grasping during initialization fails? is this reflected in the success rates presented? Is the same procedure used to initialize horizontal sliding as for vertical sliding?

2. $\textbf{Experiment scope}$: I feel the main benefit of this method is its ability to deal with large towels because of its active perception and dynamic sliding primitives. Using small towels does validate the system, but I feel is less convincing than testing on larger towels where visual methods would struggle more.
- Though large towels are experimented with (horizontal sliding), the full pipeline of grasp affordance+tactile sliding seems to never be tested on larger towels, and I wonder why this is.
- There should ideally be experiments highlighting the entire pipeline's success rate on manipulating a (small) towel from a crumpled state to folded, since this is the primary goal the paper describes, rather than reporting success on grabbing an edge and on sliding separately.
1. $\textbf{Miscellaneous notes}$:
- This paper uses towel edge sliding with tactile sensing as an evaluation, which is worth mentioning in the related work: https://arxiv.org/pdf/2011.07213.pdf
- It seems like there should be different thresholds for grasp affordance for the sim2real and finetuned datasets, since the sim2real network's confidence could be lower having never seen real towels before. However, lower confidence doesn't necessarily imply lower success (if all outputs are scaled to be lower but ordered the same way), so it would be a fairer comparison to choose a different threshold for each network using the same standardized technique (eg, based on statistics of grasp confidence on the test set)

**Summary Of Recommendation:**

I think the paper presents a compelling system for manipulating towels using a hybrid of vision+tactile perception, and should be accepted as is with changes to writing which improve clarity and some additional experiments showcasing performance on a full folding task. If full folding experiments on long towels were included, I would increase my vote to a strong accept. Right now, I would consider myself somewhere between a weak and strong accept because of the limited experiments with larger towels.

---

> ### Author Response · Authors · 2022-08-28
> **Response to Reviewer Bk2e**
>
> We thank reviewer Bk2e for their thorough feedback and address their concerns as follows:
>
> **“How was the threshold for grasp execution chosen?”**
> The threshold was chosen independently for each network and tuned to maximize performance in the offline dataset. This answer has also been incorporated into the paper.
>
> **“How is the orientation of an edge grasp determined?”**
> The cloth is rotated so that an edge is graspable, and then all grasps are executed in the same orientation. The final orientation of the edge in the grasp is then estimated with tactile feedback.  Since the success of the tactile sliding controller depends on the initial alignment of the edge, a potential improvement of the system would be to estimate the local orientation of the edge from visual feedback or regrasping the edge to achieve a better alignment.
>
> **“How is the 92% tactile classification accuracy evaluated?”**
> Tactile classification accuracy was evaluated on a held out test set evenly distributed across classes. The test set has 44 grasps per class. We have added a confusion matrix illustrating the performance per class in the appendix (Section 7.5).
>
> **“Why not use the available corner classification network for this part of the system?”**
> The corner classification network was trained with data extracted during grasping. This data has a different distribution than during sliding since there is significantly more shear and the classifier has a hard time generalizing.  In practice, also, the requirements for that initial pick of a corner are less stringent, since it is not necessary to grasp on a “clean” corner. For example the robot can pick up a corner folded in on itself and still successfully complete the task. Therefore, we found picking the lowest point and using vision to confirm that the towel is being held was sufficient for our task. Parts of this answer have been incorporated in our revision (Appendix 7.1).
>
> **“Fig 4 suggests that the grasp supervision network can detect corners”**
> The main motivation for separating the single-frame tactile sliding network and multi-frame grasp supervision network was to incorporate temporal information, while the gripper closes on a grasp, to distinguish edges from folds. This grasp supervision network can also classify corners, but the accuracy was not high enough to directly incorporate into our system (often conflated with folds). In contrast, we were able to differentiate edges and non-edges (the four other categories) with 98% accuracy.
>
> **“The success rate of corner grasping is not discussed”**
> The main failure modes of corner grasping are slightly missing the towel (because of movement while sensing or errors from the camera) or the lowest point not being a corner (if the towel is partially folded). We found the visual check is sufficient to handle the first failure case. The second case turned out to be uncommon experimentally (dependent on initial configuration).
>
> **“The initialization of experiments is unclear”**
> We choose to provide experimental results for each part of the system  individually and then demonstrate the complete system through a state machine. We test the success rates of both edge grasping and sliding with a few different initial grasps. We test 10 different hanging grasps across all baselines. Vertical sliding is tested from a consistent starting height with a manually adjusted starting edge position. Horizontal sliding is initialized by hand-feeding the towel to the robot.
>
> **“The full pipeline of grasp affordance+tactile sliding seems to never be tested on larger towels”**
> Switching to the horizontal configuration while holding onto the cloth with two grippers was a challenging maneuver with our setup. We would also likely have to retrain the grasp classification network and the visuotactile affordance network (potentially in simulation and then finetune) to demonstrate on the longer towel. Given time constraints, we demonstrate the whole system with the smaller towels.
>
> **“There should ideally be experiments highlighting the entire pipeline's success rate”**
> The original intention of this paper was to show the capabilities of visuotactile perception and control, rather than overall task success/efficiency. While each skill has a high success rate if initialized properly, the transitions require more engineering to make smooth. As addressed in the limitation section, sliding success is highly dependent on initial edge position and orientation. To address this, we could use edge position as supervision or introduce a tactile regrasp. Also, whenever an action fails, our visual and tactile checks (if accurate) can restart the state machine where appropriate. We added a system design choices section to our appendix to discuss this decision.
>
> **“This paper uses towel edge sliding with tactile sensing as an evaluation, which is worth mentioning in the related work”**
> Thanks for catching this, we have added this to our related works section.

---

### Official Review · Reviewer_a2rY · 2022-07-30

**Originality:** Very Good
**Technical Quality:** Very Good
**Clarity Of Presentation:** Excellent
**Impact:** 4

**Recommendation:**

Strong Accept: I recommend accepting the paper and will argue for my recommendation even if other reviewers hold a different opinion.

**Summary:**

The paper addresses the problem of cloth grasping by presenting a system that leverages both visual and tactile perception. The proposed system uses tactile perception (edge estimation network, tactile sliding network) to enble closed-loop local control under the occlusion induced by gripper. Another vision-based grasp affordance netowrk is pretrained in simulation and fine-tuned with tactile supervision, and provides global guidance to state identification and grasp point localization. The proposed system is evaluated in real-world and demonstrates impressive performance in real-world cloth grasping.

**Issues:**

It will be nice if the authors can provide results of the proposed method under more challenging setting (without lifting-up, clothes other than towel), or at least the discussion about it. It will help the community to better understand the limitation of the method.

**Quality Of The Limitations Section:**

Limitations are addressed clearly

**Reviewer Expertise:**

4: The reviewer is confident but not absolutely certain that the evaluation is correct

**Robotics Focus:**

Sufficient demonstration on hardware

**Strengths And Weaknesses:**

Strengths:
*  The paper tackles a very important problem in cloth manipulation: cloth grasping. When designing the system, the authors take the common failure mode of cloth grasping into account,  such as gripper collission and grasping wrong number of layers, missed grasp.

* Compared to previous cloth grasping approach, it leverages tactile perception, and in a really clever way: 1. it enables closed-loop manipulation with occlusion; 2. it provides supervision for the affordance learning.
*The proposed system outperforms all baselines in real world and achieves a success rate of 90%.


Limitations:
* The evaluation is conducted on a simplified setup.  The towel is lifted by the robot so that is partially unfolded by the gravity. I don't think edge detection on a hanging cloth is a very difficult task, as there aren't many folds or wrinkles. Many of the current cloth folding pipelines can only be applied when the cloth is placed on the table.  It would be nice to see the effectiveness of the proposed system on on-table crumpled cloth.
* The system is only evaluated for towel, which is kind of limited. I doubt it if will work for clothes with more complicated geometries, such as Tshirt (multiple corners), as double-layer clothes.


**Summary Of Recommendation:**

The paper tackles a very important but challenging problem in deformable object manipulation: cloth grasping, and an interesting approach. Accurate cloth grasping can be really helpful for the downstream tasks, such as cloth flattening and folding. I believe this paper will have a significant impact on the field and I strongly recommend for acceptance.

---

> ### Author Response · Authors · 2022-08-28
> **Response to Reviewer a2rY**
>
> We would like to thank reviewer a2RY for the thoughtful comments. We address the reviewer's concerns as follows (and have incorporated these answers into our appendix in subsections 7.1 and 7.11):
>
> **“Many of the current cloth folding pipelines can only be applied when the cloth is placed on the table. It would be nice to see the effectiveness of the proposed system on on-table crumpled cloth.”**
>
> We are particularly interested in edge grasps that allow for sliding (single layer, no collisions) with our given hardware. These grasps are usually not available for a towel resting on a table. Other papers use a foam surface or smaller, tweezer-like grippers that can more easily slide under a layer of cloth in order to grasp an edge. Our tactile grippers, while unable to directly grasp a single-layered edge from the ground due to their larger size, enable other skills (sliding, tensioning). Picking the towel off the table is an important step so that edge grasps are exposed.
>
> **“The system is only evaluated for towel, which is kind of limited. I doubt it if will work for clothes with more complicated geometries, such as Tshirt (multiple corners), as double-layer clothes.”**
>
> Tactile supervision in our framework only works if different categories (edge, corner, fold, …) can be reliably differentiated. If a feature is unique, like the hem of a shoulder strap, a zipper, or a shirt collar, then we could similarly use tactile feedback to confirm grasps of local features. However, it would be difficult to differentiate, for example, a sleeve hem and a waist hem on a T-shirt, which might be necessary for certain manipulation tasks. A reasonable way to approach this more general problem would be to use a visual prior to inform of the global structure of the cloth, as in Ganapathi et al. (2020), and use tactile sensing to confirm/refine the location of the grasp.

---

### Meta-Review · Area_Chair_DDVj · 2022-08-15

**Recommendation:** Accept (Poster)
**Confidence:** 5

**Metareview:**

The paper is of interest and addresses an important problem relating to deformable object manipulation. It utilizes a hybrid of vision and tactile perception for manipulating towels. The idea is original with very good technical quality. The work was presented well and guides the reader appropriately. The reviewer concerns were adequately addressed. Application to more complex tasks remains a concern, if further addressed, it can significantly improve the paper. However, with the current experiments, the paper makes sufficient contributions to the community.


**Best Paper Nomination:**

No

---

> ### Author Response · Authors · 2022-08-28
> **Response to Area Chair DDVj**
>
> We thank the area chair and reviewers for the thoughtful reviews. We have updated the paper and supplementary material based on the reviewer's feedback and addressed their questions below. Here are the sections we added to the appendix:
>
> * **7.1 System Design Choices.** Contains motivation and explanations for the state machine, for suspending the cloth, our usage of corner detection, and our evaluation of individual actions instead of the overall task
> * **7.2 Failure Modes.** Specifically for the actions of corner grasping, edge grasping, and transitioning to tactile sliding
> * **7.3 Network Architecture and Training Details.** For both tactile perception and affordance networks
> * **7.5 Tactile Grasp Supervision Network Results.** Included a confusion matrix
> * **7.8 Robot Implementation and Control.**
> * **7.10 Vertical Tactile Sliding.** Contains table with experimental results from different starting configurations
> * **7.12 Potential for Generalization.** Contains discussion on how our approach could be extended to other clothing types